# Random deep neural networks are biased towards simple functions

**Giacomo De Palma**
MechE & RLE
MIT
Cambridge MA 02139, USA
gdepalma@mit.edu

**Bobak T. Kiani**
MechE & RLE
MIT
Cambridge MA 02139, USA
bkiani@mit.edu

**Seth Lloyd**
MechE, Physics & RLE
MIT
Cambridge MA 02139, USA
slloyd@mit.edu

## Abstract

We prove that the binary classifiers of bit strings generated by random wide deep neural networks with ReLU activation function are biased towards simple functions. The simplicity is captured by the following two properties. For any given input bit string, the average Hamming distance of the closest input bit string with a different classification is at least $\sqrt{n/(2\pi \ln n)}$, where $n$ is the length of the string. Moreover, if the bits of the initial string are flipped randomly, the average number of flips required to change the classification grows linearly with $n$. These results are confirmed by numerical experiments on deep neural networks with two hidden layers, and settle the conjecture stating that random deep neural networks are biased towards simple functions. This conjecture was proposed and numerically explored in [Valle Pérez et al., ICLR 2019] to explain the unreasonably good generalization properties of deep learning algorithms. The probability distribution of the functions generated by random deep neural networks is a good choice for the prior probability distribution in the PAC-Bayesian generalization bounds. Our results constitute a fundamental step forward in the characterization of this distribution, therefore contributing to the understanding of the generalization properties of deep learning algorithms.

## 1 Introduction

The field of deep learning provides a broad family of algorithms to fit an unknown target function via a deep neural network and is having an enormous success in the fields of computer vision, machine learning and artificial intelligence [1–5]. The input of a deep learning algorithm is a training set, which is a set of inputs of the target function together with the corresponding outputs. The goal of the learning algorithm is to determine the parameters of the deep neural network that best reproduces the training set.

Deep learning algorithms generalize well when trained on real-world data [6]: the deep neural networks that they generate usually reproduce the target function even for inputs that are not part of the training set and do not suffer from over-fitting even if the number of parameters of the network is larger than the number of elements of the training set [7–10]. A thorough theoretical understanding of this unreasonable effectiveness is still lacking. The bounds to the generalization error of learning algorithms are proven in the probably approximately correct (PAC) learning framework [11]. Most of these bounds depend on complexity measures such as the Vapnik-Chervonenkis dimension [12, 13] or the Rademacher complexity [14, 15] which are based on the worst-case analysis and are not sufficient to explain the observed effectiveness since they become void when the number of parameters is larger than the number of training samples [10, 16–21]. A complementary approach is provided by the PAC-Bayesian generalization bounds [19, 22–25], which apply to nondeterministic learning

algorithms. These bounds depend on the Kullback-Leibler divergence [26] between the probability distribution of the function generated by the learning algorithm given the training set and an arbitrary prior probability distribution that is not allowed to depend on the training set: the smaller the divergence, the better the generalization properties of the algorithm. Making the right choice for the prior distribution is fundamental to obtain a nontrivial generalization bound.

A good choice for the prior distribution is the probability distribution of the functions generated by deep neural networks with randomly initialized weights [27]. Understanding this distribution is therefore necessary to understand the generalization properties of deep learning algorithms. PAC-Bayesian generalization bounds with this prior distribution led to the proposal that the unreasonable effectiveness of deep learning algorithms arises from the fact that the functions generated by a random deep neural network are biased towards simple functions [27–29]. Since real-world functions are usually simple [30, 31], among all the functions that are compatible with a training set made of real-world data, the simple ones are more likely to be close to the target function. The conjectured bias towards simple functions has been numerically explored in [27], which considered binary classifications of bit strings and showed that binary classifiers with a small Lempel-Ziv complexity [32] are more likely to be generated by a random deep neural network than binary classifiers with a large Lempel-Ziv complexity. However, a rigorous proof of this bias is still lacking.

## 1.1 Our contribution

We prove that random deep neural networks are biased towards simple functions, in the sense that a typical function generated is insensitive to large changes in the input. We consider random deep neural networks with Rectified Linear Unit (ReLU) activation function and weights and biases drawn from independent Gaussian probability distributions, and we employ such networks to implement binary classifiers of bit strings. Our main results are the following:

- We prove that for $n \gg 1$, where $n$ is the length of the string, for any given input bit string the average Hamming distance of the closest bit string with a different classification is at least $\sqrt{n/(2\pi \ln n)}$ (Theorem 1), where the Hamming distance between two bit strings is the number of different bits.

- We prove that, if the bits of the initial string are randomly flipped, the average number of bit flips required to change the classification grows linearly with $n$ (Theorem 2). From a heuristic argument, we find that the average required number of bit flips is at least $n/4$ (subsection 3.3), and simulations on deep neural networks with two hidden layers indicate a scaling of approximately $n/3$.

By contrast, for a random binary classifier drawn from the uniform distribution over all the possible binary classifiers of strings of $n \gg 1$ bits, the average Hamming distance of the closest bit string with a different classification is one, and the average number of random bit flips required to change the classification is two. Therefore, our result identifies a fundamental qualitative difference between a typical binary classifier generated by a random deep neural network and a uniformly random binary classifier.

The result proves that the binary classifiers generated by random deep neural networks are simple and identifies the classifiers that are likely to be generated as the ones with the property that a large number of bits need to be flipped in order to change the classification. While all the classifiers with this property have a low Kolmogorov complexity[1], the converse is not true. For example, the parity function has a small Kolmogorov complexity, but it is sufficient to flip just one bit of the input to change the classification, hence our result implies that it occurs with a probability exponentially small in $n$. Similarly, our results explain why [27] found that the look-up tables for the functions generated by random deep networks are typically highly compressible using the LZW algorithm [35], which identifies statistical regularities, but not all functions with highly compressible look-up tables are likely to be generated.

The proofs of Theorems 1 and 2 are based on the approximation of random deep neural networks as Gaussian processes, which becomes exact in the limit of infinite width [36–47]. The crucial property of random deep neural networks captured by this approximation is that the outputs generated by

inputs whose Hamming distance grows sub-linearly with $n$ become perfectly correlated in the limit $n \to \infty$. These strong correlations are the reason why a large number of input bits need to be flipped in order to change the classification. The proof of Theorem 2 also exploits the theory of stochastic processes, and in particular the Kolmogorov continuity theorem [48]. We stress that for activation functions other than the ReLU, the scaling with $n$ of both the Hamming distance of the closest bit string with a different classification and the number of random bit flips necessary to change the classification remain the same. However, the prefactor can change and can be exponentially small in the number of hidden layers.

We validate all the theoretical results with numerical experiments on deep neural networks with ReLU activation function and two hidden layers. The experiments confirm the scalings $\Theta(\sqrt{n/\ln n})$ and $\Theta(n)$ for the Hamming distance of the closest string with a different classification and for the average random flips required to change the classification, respectively. The theoretical pre-factor $1/\sqrt{2\pi}$ for the closest string with a different classification is confirmed within an extremely small error of $1.5\%$. The heuristic argument that pre-factor for the random flips is greater than $1/4$ is confirmed by numerics which indicate that the pre-factor is approximately $0.33$. Moreover, we explore the Hamming distance to the closest bit string with a different classification on deep neural networks trained on the MNIST database [49] of hand-written digits. The experiments show that the scaling $\Theta(\sqrt{n/\ln n})$ survives after the training of the network and that the distance of a training or test picture from the closest classification boundary is strongly correlated with its classification accuracy, i.e., the correctly classified pictures are further from the boundary than the incorrectly classified ones.

## 1.2 Further related works

The properties of deep neural networks with randomly initialized weights have been the subject of intensive studies [38–42, 50–52]. The relation between generalization and simplicity for Boolean function was explored in [53], where the authors provide numerical evidence that the generalization error is correlated with a complexity measure that they define. Ref. [10] explores the generalization properties of deep neural networks trained on partially random data, and finds that the generalization error correlates with the amount of randomness in the data. Based on this result, Ref. [28,54] proposed that the stochastic gradient descent employed to train the network is more likely to find the simpler functions that match the training set rather than the more complex ones. However, further studies [29] suggested that stochastic gradient descent is not sufficient to justify the observed generalization. The idea of a bias towards simple patterns has been applied to learning theory through the concepts of minimum description length [55], Blumer algorithms [56, 57] and universal induction [34]. Ref. [58] proved that the generalization error grows with the Kolmogorov complexity of the target function if the learning algorithm returns the function that has the lowest Kolmogorov complexity among all the functions compatible with the training set. The relation between generalization and complexity has been further investigated in [30, 59]. The complexity of the functions generated by a deep neural networks has also been studied from the perspective of the number of linear regions [60–62] and of the curvature of the classification boundaries [41]. We note that the results proved here — viz., that the functions generated by random deep networks are insensitive to large changes in their inputs — implies that such functions should be simple with respect to all the measures of complexity above, but the converse is not true: not all simple functions are likely to be generated by random deep networks.

## 2 Setup and Gaussian process approximation

We consider a feed-forward deep neural network with $L$ hidden layers, activation function $\tau$, input in $\mathbb{R}^n$ and output in $\mathbb{R}$. The most common choice for $\tau$ is the ReLU activation function $\tau(x) = \max(0, x)$. We stress that Theorems 1 and 2 do not rely on this assumption and hold for any activation function. For any $x \in \mathbb{R}^n$ and $l = 2, \ldots, L+1$, the network is recursively defined by

$$\phi^{(1)}(x) = W^{(1)}x + b^{(1)}, \qquad \phi^{(l)}(x) = W^{(l)}\tau\left(\phi^{(l-1)}(x)\right) + b^{(l)}, \tag{1}$$

where $\phi^{(l)}(x)$, $b^{(l)} \in \mathbb{R}^{n_l}$, $W^{(l)}$ is an $n_l \times n_{l-1}$ real matrix, $n_0 = n$ and $n_{L+1} = 1$. We put for simplicity $\phi = \phi^{(L+1)}$, and we define $\psi(x) = \text{sign}\left(\phi(x)\right)$ for any $x \in \mathbb{R}^n$. The function $\psi$ is a binary classifier on the set of the strings of $n$ bits identified with the set $\{-1, 1\}^n \subset \mathbb{R}^n$, where the classification of the string $x \in \{-1, 1\}^n$ is $\psi(x) \in \{-1, 1\}$. We choose this representation of the bit strings since any $x \in \{-1, 1\}^n$ has $\|x\|^2 = n$, and the covariance of the Gaussian process

approximating the deep neural network has a significantly simpler expression if all the inputs have the same norm. Moreover, having the inputs lying on a sphere is a common assumption in the machine learning literature [63].

We draw each entry of each $W^{(l)}$ and of each $b^{(l)}$ from independent Gaussian distributions with zero mean and variances $\sigma_w^2/n_{l-1}$ and $\sigma_b^2$, respectively. We employ the Gaussian process approximation of [41, 42], which consists in assuming that for any $l$ and any $x, y \in \mathbb{R}^n$, the joint probability distribution of $\phi^{(l)}(x)$ and $\phi^{(l)}(y)$ is Gaussian, and $\phi_i^{(l)}(x)$ is independent from $\phi_j^{(l)}(y)$ for any $i \neq j$. This approximation is exact for $l = 1$ and holds for any $l$ in the limit $n_1, \ldots, n_L \to \infty$ [39]. Indeed, $\phi_i^{(l)}(x)$ is the sum of $b_i^{(l)}$, which has a Gaussian distribution, with the $n_{l-1}$ terms $\{W_{ij}^{(l)} \tau(\phi_j^{(l-1)}(x))\}_{j=1}^{n_{l-1}}$ which are iid from the inductive hypothesis. Therefore if $n_{l-1} \gg 1$, from the central limit theorem $\phi_i^{(l)}(x)$ has a Gaussian distribution. We notice that for finite width, the outputs of the intermediate layers have a sub-Weibull distribution [64]. Our experiments in section 4 show agreement with the Gaussian approximation starting from $n \gtrsim 100$.

In the Gaussian process approximation, for any $x, y$ with $\|x\|^2 = \|y\|^2 = n$, the joint probability distribution of $\phi(x)$ and $\phi(y)$ is Gaussian with zero mean and covariance that depends on $x, y$ and $n$ only through $x \cdot y / n$:

$$\mathbb{E}(\phi(x)) = 0, \qquad \mathbb{E}(\phi(x)\,\phi(y)) = Q\,F\left(\tfrac{x \cdot y}{n}\right), \qquad \|x\|^2 = \|y\|^2 = n. \tag{2}$$

Analogously, $\phi(x)$ is a Gaussian process with zero average and covariance given by the kernel $K(x, y) = Q\,F\left(\tfrac{x \cdot y}{n}\right)$. Here $Q > 0$ is a suitable constant and $F : [-1, 1] \to \mathbb{R}$ is a suitable function that depend on $\tau, L, \sigma_w$ and $\sigma_b$, but not on $n$, $x$ nor $y$. We have introduced the constant $Q$ because it will be useful to have $F$ satisfy $F(1) = 1$. We provide the expression of $Q$ and $F$ in terms of $\tau, L$, $\sigma_w$ and $\sigma_b$ in the supplementary material, where we also prove that for the ReLU activation function $t \leq F(t) \leq 1$.

The correlations between outputs of the network generated by close inputs are captured by the behavior of $F(t)$ for $t \to 1$. If $F(t)$ stays close to $1$ as $t$ departs from $1$, then the outputs generated by close inputs are almost perfectly correlated and have the same classification with probability close to one. On the contrary, if $F(t)$ drops quickly, the correlations decay and there is a nonzero probability that close inputs have different classifications. In the supplementary material we prove that for the ReLU activation function we have $0 < F'(1) \leq 1$ and for $t \to 1$,

$$F(t) = 1 - F'(1)\,(1 - t) + O\left((1 - t)^{\frac{3}{2}}\right), \tag{3}$$

implying strong short-distance correlations.

## 3 Theoretical results

### 3.1 Closest bit string with a different classification

Our first main result is the following Theorem 1, which states that for $n \gg 1$, for any given input bit string of a random deep neural network as in section 2 the average Hamming distance of the closest input bit string with a different classification is $\sqrt{n/(2\pi F'(1)\ln n)}$. The proof is in the supplementary material.

**Theorem 1** (closest string with a different classification). *For any $n \in \mathbb{N}$, let $\phi : \{-1, 1\}^n \to \mathbb{R}$ be the output of a random deep neural network as in section 2. Let $a > 0$ and let $h_n = \lfloor a\sqrt{n/\ln n} \rfloor$, where $\lfloor t \rfloor$ denotes the integer part of $t \geq 0$. Let us fix $x \in \{-1, 1\}^n$ and $z > 0$, and let $N_n(a, z)$ be the average number of input bit strings $y \in \{-1, 1\}^n$ with Hamming distance $h_n$ from $x$ and with a different classification from $x$, conditioned on $\phi(x) = \sqrt{Q}\,z$:*

$$N_n(a, z) = \mathbb{E}\left(\# \left\{y \in \{-1, 1\}^n : h(x, y) = h_n, \phi(y) < 0\right\} \Big| \phi(x) = \sqrt{Q}\,z\right). \tag{4}$$

*Here $h(x, y)$ is the Hamming distance between $x$ and $y$ and we recall that $Q = \mathbb{E}(\phi(x)^2)$. Then, for $n \to \infty$*

$$\ln N_n(a, z) = \frac{a}{2}\sqrt{n \ln n}\left(1 - \frac{z^2}{4F'(1)a^2} + \frac{\ln \frac{\ln n}{a^2}}{\ln n} + O\left(\frac{1}{\sqrt[4]{n \ln n}}\right)\right). \tag{5}$$

*In particular,*

$$\lim_{n\to\infty} N_n(a,z) = 0 \quad \text{for} \quad a < \frac{z}{2\sqrt{F'(1)}}, \qquad \lim_{n\to\infty} N_n(a,z) = \infty \quad \text{for} \quad a \ge \frac{z}{2\sqrt{F'(1)}}. \quad (6)$$

Theorem 1 tells us that, if $n \gg 1$, for any input bit string $x \in \{-1,1\}^n$, with very high probability all the input bit strings $y \in \{-1,1\}^n$ with Hamming distance from $x$ lower than

$$h_n^*(x) = \frac{|\phi(x)|}{2\sqrt{Q\,F'(1)}} \sqrt{\frac{n}{\ln n}} \quad (7)$$

have the same classification as $x$, i.e., $\phi(y)$ has the same sign as $\phi(x)$. Moreover, the number of input bit strings $y$ with Hamming distance from $x$ higher than $h_n^*(x)$ and with a different classification than $x$ is exponentially large in $n$. Therefore, with very high probability the Hamming distance from $x$ of the closest bit string with a different classification is approximately $h_n^*(x)$. Since $\mathbb{E}(|\phi(x)|) = \sqrt{2Q/\pi}$, the average Hamming distance of the closest string with a different classification is

$$\mathbb{E}\left(h_n^*(x)\right) = \sqrt{\frac{n}{2\pi F'(1)\ln n}} \ge \sqrt{\frac{n}{2\pi \ln n}}, \quad (8)$$

where the last inequality holds for the ReLU activation function and follows since in this case $F'(1) \le 1$.

*Remark* 1. While Theorem 1 holds for any activation function, the property $F'(1) \le 1$ may not hold for activation functions different from the ReLU. For example, in the case of $\tanh$ there are values of $\sigma_w$ and $\sigma_b$ such that $F'(1)$ grows exponentially with $L$ [41]. In this case, the Hamming distance of the closest string with a different classification still scales as $\sqrt{n/\ln n}$, but the prefactor can be exponentially small in $L$. Therefore with the $\tanh$ activation function, for finite values of $L$ and $n$, $F'(1)$ may become comparable with $\sqrt{n/\ln n}$ and significantly affect the Hamming distance.

### 3.2 Random bit flips

Let us now consider the problem of the average number of bits that are needed to flip in order to change the classification of a given bit string. We consider a random sequence of input bit strings $\{x^{(0)}, \ldots, x^{(n)}\} \subset \{-1,1\}^n$, where at the $i$-th step $x^{(i)}$ is generated flipping a random bit of $x^{(i-1)}$ that has not been already flipped in the previous steps. Any sequence as above is geodesic, i.e., $h(x^{(i)}, x^{(j)}) = |i - j|$ for any $i, j = 0, \ldots, n$. The following Theorem 2 states that the average Hamming distance from $x^{(0)}$ of the closest string of the sequence with a different classification is proportional to $n$. The proof is in the supplementary material.

**Theorem 2** (random bit flips). *For any $n \in \mathbb{N}$, let $\phi : \{-1,1\}^n \to \mathbb{R}$ be the output of a random deep neural network as in section 2, and let $\{x^{(0)}, \ldots, x^{(n)}\} \subset \{-1,1\}^n$ be a geodesic sequence of bit strings. Let $h_n$ be the expected value of the minimum number of steps required to reach a bit string with a different classification from $x^{(0)}$:*

$$h_n = \mathbb{E}\left(\min\left\{\min\left\{1 \le i \le n : \phi(x^{(0)})\phi(x^{(i)}) < 0\right\}, n\right\}\right). \quad (9)$$

*Then, there exists a constant $t_0 > 0$ which depends only on $F$ such that $h_n \ge n\,t_0$ for any $n \in \mathbb{N}$.*

*Remark* 2. Since the entry of the kernel (2) associated to two inputs lying on the sphere is a function of their squared Euclidean distance, which coincides with the Hamming distance in the case of bit strings, Theorems 1 and 2 may be generalized to continuous inputs on the sphere by replacing the Hamming distance with the squared Euclidean distance.

### 3.3 Heuristic argument

For a better understanding of Theorems 1 and 2, we provide a simple heuristic argument to their validity. The crucial observation is that, if one bit of the input is flipped, the change in $\phi$ is $\Theta(1/\sqrt{n})$. Indeed, let $x, y \in \{-1,1\}^n$ with $h(x,y) = 1$. From (2), $\phi(y) - \phi(x)$ is a Gaussian random variable with zero average and variance

$$\mathbb{E}\left((\phi(y) - \phi(x))^2\right) = 2Q\left(1 - F\left(1 - \tfrac{2}{n}\right)\right) \simeq 4QF'(1)/n. \quad (10)$$

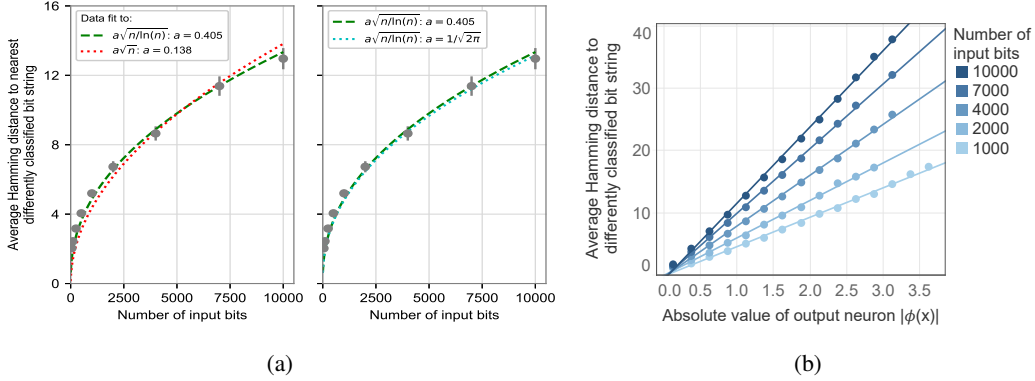

(a)                                                                    (b)

Figure 1: (a) Average Hamming distance to the nearest differently classified input string versus the number of input neurons for the neural network. The Hamming distance to the nearest differently classified string scales as $\sqrt{n/(2\pi \ln n)}$ with respect to the number of input neurons. Left: the results of the simulations clearly show the importance of the $\ln n$ term in the scaling. Right: the empirically calculated value $0.405$ for the pre-factor $a$ is close to the theoretically predicted value of $1/\sqrt{2\pi}$. Each data point is the average of 1000 different calculations of the Hamming distance for randomly sampled bit strings. Each calculation was performed on a randomly generated neural network. Further technical details for the design of the neural networks are given in subsection 4.4. (b) The linear relationship between $|\phi(x)|$ and $h_n^*(x)$ is consistent across neural networks of different sizes. To calculate the average distance at values of $|\phi(x)|$ within an interval, data was averaged across equally spaced bins of 0.25 for values of $|\phi(x)|$. Averages for each bin are plotted at the midpoint of the bin. Points are only shown if there are at least 10 samples within the bin.

For any $i$, at the $i$-th step of the sequence of bit strings of subsection 3.2, $\phi$ changes by the Gaussian random variable $\phi(x^{(i)}) - \phi(x^{(i+1)})$, which from (10) has zero mean and variance $4Q\,F'(1)/n$. Assuming that the changes are independent, after $h$ steps $\phi$ changes by a Gaussian random variable with zero mean and variance $4h\,Q\,F'(1)/n$. Recalling that $\mathbb{E}(\phi(x^{(0)})^2) = Q$ and that $F'(1) \leq 1$ for the ReLU activation function, approximately $h \approx n/(4F'(1)) \geq n/4$ steps are needed in order to flip the sign of $\phi$ and hence the classification.

Let us now consider the problem of the closest bit string with a different classification from a given bit string $x$. For any bit string $y$ at Hamming distance one from $x$, $\phi(y) - \phi(x)$ is a Gaussian random variable with zero mean and variance $4Q\,F'(1)/n$. We assume that these random variables are independent, and recall that the minimum among $n$ iid normal Gaussian random variables scales as $\sqrt{2\ln n}$ [65]. There are $n$ bit strings $y$ at Hamming distance one from $x$, therefore the minimum over these $y$ of $\phi(y) - \phi(x)$ is approximately $-\sqrt{8Q\,F'(1)\ln n/n}$. This is the maximum amount by which we can decrease $\phi$ flipping one bit of the input. Iterating the procedure, the maximum amount by which we can decrease $\phi$ flipping $h$ bits is $h\sqrt{8Q\,F'(1)\ln n/n}$. Since $\mathbb{E}(\phi(x^{(0)})^2) = Q$, the minimum number of bit flips required to flip the sign of $\phi$ is approximately $h \approx \sqrt{n/(8F'(1)\ln n)} \geq \sqrt{n/(8\ln n)}$, where the last inequality holds for the ReLU activation function. The pre-factor $1/\sqrt{8} \simeq 0.354$ obtained with the heuristic proof above is very close to the exact pre-factor $1/\sqrt{2\pi} \simeq 0.399$ obtained with the formal proof in (8).

## 4 Experiments

### 4.1 Closest bit string with a different classification

To confirm experimentally the findings of Theorem 1, Hamming distances to the closest bit string with a different classification were calculated for randomly generated neural networks with parameters sampled from normal distributions (see subsection 4.4). This distance was calculated using a greedy search algorithm (Figure 1a). In this algorithm, the search for a differently classified bit string progressed in steps, where in each step, the most significant bit was flipped. This bit corresponded to the one that produced the largest change towards zero in the value of the output neuron when

flipped. To ensure that this algorithm accurately calculated Hamming distances, we compared the results of the greedy search algorithm to those from an exact search which exhaustively searched all bit strings at specified Hamming distances for smaller networks where this exact search method was computationally feasible. Comparisons between the two algorithms in Table 1 of the supplementary material show that outcomes from the greedy search algorithm were consistent with those from the exact search algorithm. The results from the greedy search method confirm the $\sqrt{n/\ln n}$ scaling of the average Hamming distance starting from $n \gtrsim 100$. The value of the pre-factor $1/\sqrt{2\pi}$ is also confirmed with the high precision of $1.5\%$. Figure 1b empirically validates the linear relationship between the value of the output neuron $|\phi(x)|$ and the Hamming distance to bit strings with different classification $h_n^*(x)$ expressed by (7). This linear relationship was consistent with all neural networks empirically tested in our analysis. Intuitively, $|\phi(x)|$ is an indication of the confidence in classification. The linear relationship shown here implies that as the value of $|\phi(x)|$ grows, the confidence of the classification of an input strengthens, increasing the distance from that input to boundaries of different classifications.

## 4.2 Random bit flips

Figure 2 confirms the findings of Theorem 2, namely that the expected number of random bit flips required to reach a bit string with a different classification scales linearly with the number of input neurons. The pre-factor found by simulation is $0.33$, slightly above the lower bound of $0.25$ estimated from the heuristic argument. Our results show that, though the Hamming distance to the nearest classification boundary scales on average at a rate of $\sqrt{n/\ln n}$, the distance to a random boundary scales linearly and more rapidly.

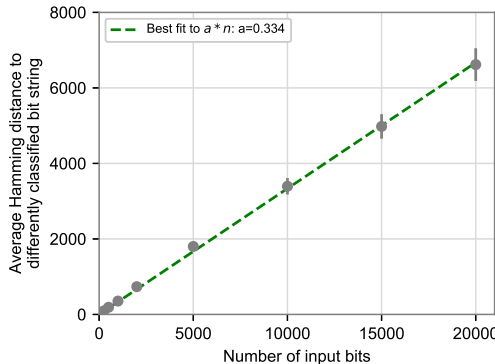

Figure 2: The average number of random bit flips required to reach a bit string with different classification scales linearly with the number of input neurons. Each point is averaged across a sample of 1000 neural networks, where the Hamming distances to differently classified bit strings for each network are tested at a single random input bit string.

## 4.3 Analysis of MNIST data

Our theoretical results hold for random, untrained deep neural networks. It is an interesting question whether trained deep neural networks exhibit similar properties for the Hamming distances to classification boundaries. Clearly some trained networks will not: a network that has been trained to return as output the final bit of the input string has Hamming distance one to the nearest classification boundary. For networks that are trained to classify noisy data, however, we expect the trained networks to exhibit relatively large Hamming distances to the nearest classification boundary. Moreover, if a 'typical' network can perform the noisy classification task, then we expect training to guide the weights to a nearby typical network that does the job, for the simple reason that networks that exhibit $\Theta(\sqrt{n/\ln n})$ distance to the nearest boundary and an average distance of $\Theta(n)$ to a boundary under random bit flips have much higher prior probabilities than atypical networks.

To determine if our results hold for models trained on real-world data, we trained 2-layer fully-connected neural networks to categorize whether hand-drawn digits taken from the MNIST database [66] are even or odd. Images of hand drawn digits were converted from their 2-dimensional format (28 by 28 pixels) into a 1-dimensional vector of 784 binary inputs. The starting 8 bit pixel values were converted to binary format by determining whether the pixel value was above or below a threshold of 25. Networks were trained to determine whether the hand-drawn digit was odd or even. All networks

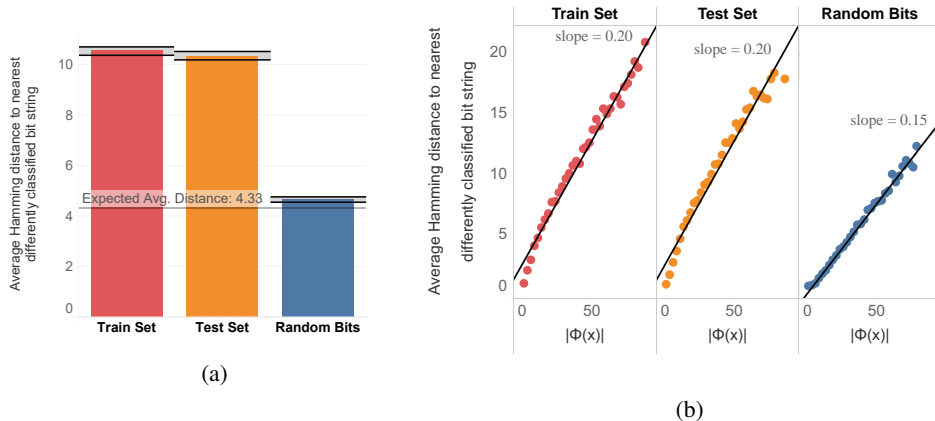

(a)

(b)

Figure 3: (a) Average Hamming distance to the nearest differently classified input bit string for MNIST trained models calculated using the greedy search method. The average distance calculated for random bits is close to the expected value of approximately 4.33. Further technical details for the design of the neural networks are given in subsection 4.4.
(b) The linear relationship between $|\phi(x)|$ and $h_n^*(x)$ is consistent for networks trained on MNIST data. To calculate the average distance at values of $|\phi(x)|$ within an interval, data was averaged across equally spaced bins of 2.5 for values of $|\phi(x)|$. Averages for each bin are plotted at the midpoint of the bin. Points are only shown if there are at least 25 samples within the bin.

followed the design described in subsection 4.4. 400 Networks were trained for 20 epochs using the Adam optimizer [67]; average test set accuracy of 98.8% was achieved.

For these trained networks, Hamming distances to the nearest bit string with a different classification were calculated using the greedy search method outlined in subsection 4.1. These Hamming distances were evaluated for three types of bit strings: bit strings taken from the training set, bit strings taken from the test set, and randomly sampled bit strings where each bit has equal probability of 0 and 1. For the randomly sampled bit strings, the average minimum Hamming distance to a differently classified bit string is very close to the expected theoretical value of $\sqrt{n/(2\pi \ln n)}$ (Figure 3a). By contrast, for bit strings taken from the test and training set, the minimum Hamming distances to a classification boundary were on average much higher than that for random bits, as should be expected: training increases the distance from the data points to the boundary of their respective classification regions and makes the network more robust to errors when classifying real-world data compared with classifying random bit strings.

Furthermore, even for trained networks, a linear relationship is still observed between the absolute value of the output neuron (prior to normalization by a sigmoid activation) and the average Hamming distance to the nearest differently classified bit string (Figure 3b). Here, the slope of the linear relationship is larger for test and training set data, consistent with the expectation that training should extend the Hamming distance to classification boundaries for patterns of data found in the training set.

Finally, we have explored the correlation between the distance of a training or test picture from the closest classification boundary with its classification accuracy. Figure 4 shows that the incorrectly classified pictures tend to be significantly closer to the classification boundary than the correctly classified ones: the average distances are 1.42 and 10.61, respectively, for the training set, and 2.30 and 10.47, respectively, for the test set. Therefore, our results show that the distance to the closest classification boundary is empirically correlated with the classification accuracy and with the generalization properties of the deep neural network.

## 4.4 Experimental apparatus and structure of neural networks

Weights for all neural networks are initialized according to a normal distribution with zero mean and variance equal to $2/n_{in}$, where $n_{in}$ is the number of input units in the weight tensor. No bias term is included in the neural networks. All networks consist of two fully connected hidden layers,

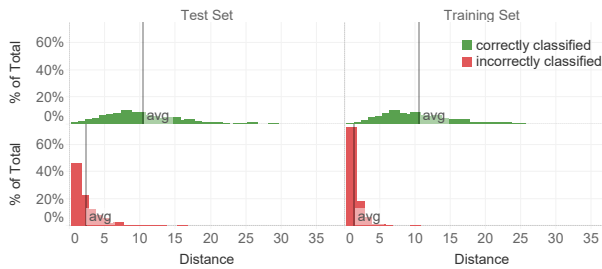

Figure 4: Histogram counting instances of correctly and incorrectly classified MNIST pictures shows that trained neural networks are far more likely to misclassify points closer to a classification boundary for both the training and test sets. Results are aggregated across 20 different trained neural networks trained to classify whether digits are even or odd. Networks are trained for 10 epochs using the Adam optimizer.

each with $n$ neurons (equal to number of input neurons) and activation function set to the commonly used Rectified Linear Unit (ReLU). All networks contain a single output neuron with no activation function. In the notation of section 2, this choice corresponds to $\sigma_w^2 = 2$, $\sigma_b^2 = 0$, $n_0 = n_1 = n_2 = n$ and $n_3 = 1$, and implies $F'(1) = 1$. Simulations were run using the python package Keras with a backend of TensorFlow [68].

## 5 Conclusions

We have proved that the binary classifiers of strings of $n \gg 1$ bits generated by wide random deep neural networks with ReLU activation function are simple. The simplicity is captured by the following two properties. First, for any given input bit string the average Hamming distance of the closest input bit string with a different classification is at least $\sqrt{n/(2\pi \ln n)}$. Second, if the bits of the original string are randomly flipped, the average number of bit flips needed to change the classification is at least $n/4$. For activation functions other than the ReLU both scalings remain the same, but the prefactor can change and can be exponentially small in the number of hidden layers.

The striking consequence of our result is that the binary classifiers of strings of $n \gg 1$ bits generated by a random deep neural network lie with very high probability in a subset which is an exponentially small fraction of all the possible binary classifiers. Indeed, for a uniformly random binary classifier, the average Hamming distance of the closest input bit string with a different classification is one, and the average number of bit flips required to change the classification is two. Our result constitutes a fundamental step forward in the characterization of the probability distribution of the functions generated by random deep neural networks, which is employed as prior distribution in the PAC-Bayesian generalization bounds. Therefore, our result can contribute to the understanding of the generalization properties of deep learning algorithms.

Our analysis of the MNIST data suggests that, for certain types of problems, the property that many bits need to be flipped in order to change the classification survives after training the network. Both our theoretical results and our experiments are completely consistent to the empirical findings in the context of adversarial perturbations [69–74], where the existence of inputs that are close to a correctly classified input but have the wrong classification is explored. As expected, our results show that as the size of the input grows, the average number of bits needed to be flipped to change the classification increases in absolute terms but decreases as a percentage of the total number of bits. An extension of our theoretical results to trained deep neural networks would provide a fundamental robustness result of deep neural networks with respect to adversarial perturbations, and will be the subject of future work.

Moreover, our experiments on MNIST show that the distance of a picture to the closest classification boundary is correlated with its classification accuracy and thus with the generalization properties of deep neural networks, and confirm that exploring the properties of this distance is a promising route towards proving the unreasonably good generalization properties of deep neural networks.

Finally, the simplicity bias proven in this paper might shed new light on the unexpected empirical property of deep learning algorithms that the optimization over the network parameters does not suffer from bad local minima, despite the huge number of parameters and the non-convexity of the function to be optimized [75–79].

## Acknowledgements

GdP thanks the Research Laboratory of Electronics of the Massachusetts Institute of Technology for the kind hospitality in Cambridge, and Dario Trevisan for useful discussions.

GdP acknowledges financial support from the European Research Council (ERC Grant Agreements Nos. 337603 and 321029), the Danish Council for Independent Research (Sapere Aude), VILLUM FONDEN via the QMATH Centre of Excellence (Grant No. 10059) and AFOSR and ARO under the Blue Sky program. SL and BTK were supported by IARPA, NSF, BMW under the MIT Energy Initiative, and ARO under the Blue Sky program.

## Footnotes

[1]The Kolmogorov complexity of a function is the length of the shortest program that implements the function on a Turing machine [26, 33, 34].

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
