[Supplementary Material]

# Random deep neural networks are biased towards simple functions: supplementary material

**Giacomo De Palma**
MechE & RLE
MIT
Cambridge MA 02139, USA
gdepalma@mit.edu

**Bobak T. Kiani**
MechE & RLE
MIT
Cambridge MA 02139, USA
bkiani@mit.edu

**Seth Lloyd**
MechE, Physics & RLE
MIT
Cambridge MA 02139, USA
slloyd@mit.edu

## 1 Setup and Gaussian process approximation

We consider a feed-forward deep neural network with $L$ hidden layers, activation function $\tau$, input in $\mathbb{R}^n$ and output in $\mathbb{R}$. For any $x \in \mathbb{R}^n$ and $l = 2, \ldots, L+1$, the network is recursively defined by

$$\phi^{(1)}(x) = W^{(1)} x + b^{(1)}, \qquad \phi^{(l)}(x) = W^{(l)} \tau\left(\phi^{(l-1)}(x)\right) + b^{(l)}, \qquad x \in \mathbb{R}^n, \qquad (1)$$

where $\phi^{(l)}(x)$, $b^{(l)} \in \mathbb{R}^{n_l}$, $W^{(l)}$ is an $n_l \times n_{l-1}$ real matrix, $n_0 = n$ and $n_{L+1} = 1$. We put for simplicity $\phi = \phi^{(L+1)}$.

We draw each entry of each $W^{(l)}$ and of each $b^{(l)}$ from independent Gaussian distributions with zero mean and variances $\sigma_w^2/n_{l-1}$ and $\sigma_b^2$, respectively. This implies for any $x$, $y \in \mathbb{R}^n$

$$\mathbb{E}\left(\phi^{(l)}(x)\right) = 0, \quad \mathbb{E}\left(\phi_i^{(l)}(x)\, \phi_j^{(l)}(y)\right) = \delta_{ij}\, G_l(x, y). \qquad (2)$$

We determine the covariance function $G_l$ in the Gaussian process approximation of [1, 2], which consists in assuming that for any $l$ and any $x$, $y \in \mathbb{R}^n$, the joint probability distribution of $\phi^{(l)}(x)$ and $\phi^{(l)}(y)$ is Gaussian.

We start with the diagonal elements $G_l(x, x)$, which depend on $x$ and $n$ only through $\|x\|^2/n$ [1]. Since any $x \in \{-1, 1\}^n$ has $\|x\|^2 = n$, we put by simplicity for any $x \in \mathbb{R}^n$ with $\|x\|^2 = n$

$$G_l(x, x) = Q_l. \qquad (3)$$

The constants $Q_l$ can be computed from the recursive relation [1]

$$Q_1 = \sigma_w^2 + \sigma_b^2, \qquad Q_l = \sigma_w^2 \int_{-\infty}^{\infty} \tau\left(\sqrt{Q_{l-1}}\, z\right)^2 e^{-\frac{z^2}{2}} \frac{\mathrm{d}z}{\sqrt{2\pi}} + \sigma_b^2. \qquad (4)$$

We now consider the off-diagonal elements of $G_l$. For $\|x\|^2 = \|y\|^2 = n$, the correlation coefficients

$$C_l(x, y) = \frac{G_l(x, y)}{Q_l} \qquad (5)$$

depend on $x$, $y$ and $n$ only through the combination $x \cdot y/n$ [1]. We can therefore put

$$C_l(x, y) = F_l\left(\frac{x \cdot y}{n}\right), \quad \|x\|^2 = \|y\|^2 = n. \qquad (6)$$

The functions $F_l : [-1, 1] \to \mathbb{R}$ satisfy by definition $F_l(1) = 1$ and can be computed from the recursive relation [1]

$$F_1(t) = \frac{\sigma_w^2\, t + \sigma_b^2}{\sigma_w^2 + \sigma_b^2},$$

$$Q_l F_l(t) = \sigma_w^2 \int_{\mathbb{R}^2} \tau\left(\sqrt{Q_{l-1}}\, z\right) \tau\left(\sqrt{Q_{l-1}}\left(F_{l-1}(t) z + \sqrt{1 - F_{l-1}(t)^2}\, w\right)\right) e^{-\frac{z^2 + w^2}{2}} \frac{\mathrm{d}z\,\mathrm{d}w}{2\pi}$$

$$+ \sigma_b^2. \qquad (7)$$

Defining $F = F_{L+1}$ and $Q = Q_{L+1}$, the covariance of the function $\phi$ generated by the deep neural network is

$$\mathbb{E}\left(\phi(x)\,\phi(y)\right) = Q\,F\left(\frac{x\cdot y}{n}\right).\tag{8}$$

For the ReLU activation function, (7) simplifies to [3]

$$F_l(t) = \frac{Q_{l-1}\,\sigma_w^2\,\Psi\left(F_{l-1}(t)\right) + 2\sigma_b^2}{Q_{l-1}\,\sigma_w^2 + 2\sigma_b^2},\tag{9}$$

where

$$\Psi(t) = \frac{\sqrt{1-t^2} + (\pi - \arccos t)\,t}{\pi}.\tag{10}$$

The function $\Psi$ satisfies for $t \to 1$

$$\Psi(t) = t + O\left((1-t)^{\frac{3}{2}}\right).\tag{11}$$

**Proposition 1.** *For the ReLU activation function, $t \le F(t) \le 1$ for any $-1 \le t \le 1$.*

*Proof.* We prove by induction that $t \le F_l(t) \le 1$. From (7), the claim is true for $l = 1$. Let us assume the claim for $l - 1$. We have

$$\Psi'(t) = 1 - \frac{\arccos t}{\pi} \ge 0,\tag{12}$$

hence $\Psi$ is increasing. We also have $\Psi'(t) \le 1$ and $\Psi(1) = 1$, hence $\Psi(t) \ge t$. Finally, we have from (9) and from the inductive hypothesis

$$F_l(t) \ge \Psi\left(F_{l-1}(t)\right) \ge \Psi(t) \ge t,\tag{13}$$

and the claim for $l$ follows. $\qquad\square$

**Proposition 2** (short-distance correlations)**.** *For the ReLU activation function,*

$$F(t) = 1 - F'(1)\,(1-t) + O\left((1-t)^{\frac{3}{2}}\right)\tag{14}$$

*for $t \to 1$, where $F'(1)$ is determined by the recursive relation*

$$F_1'(1) = \frac{\sigma_w^2}{\sigma_w^2 + \sigma_b^2},\qquad F_l'(1) = \frac{Q_{l-1}\,\sigma_w^2}{Q_{l-1}\,\sigma_w^2 + 2\sigma_b^2}\,F_{l-1}'(1),\qquad F'(1) = F_{L+1}'(1)\tag{15}$$

*and satisfies $0 < F'(1) \le 1$.*

*Proof.* The recursive relation (15) follows taking the derivative of (9) in $t = 1$. Eq. (15) implies $0 < F_l'(1) \le 1$ for any $l$, hence $0 < F'(1) \le 1$.

The claim in (14) follows if we prove by induction that

$$F_l(t) = 1 - F_l'(1)\,(1-t) + O\left((1-t)^{\frac{3}{2}}\right)\tag{16}$$

for any $l$. The claim is true for $l = 1$. Let us assume by induction (16) for $l - 1$. We have from (11)

$$\Psi\left(F_{l-1}(t)\right) = 1 - F_{l-1}'(1)\,(1-t) + O\left((1-t)^{\frac{3}{2}}\right),\tag{17}$$

and the claim (16) for $l$ follows from (9) and (15). $\qquad\square$

## 2 Proof of Theorem 1

Let $x, y \in \{-1, 1\}^n$ with $h(x, y) = h_n$. From (8) we get $\mathbb{E}(\phi(x)\,\phi(y)) = Q\,F(1 - \frac{2h_n}{n})$, then

$$\mathbb{E}\left(\phi(y)\,\middle|\,\phi(x) = \sqrt{Q}\,z\right) = F\left(1 - \tfrac{2h_n}{n}\right)\sqrt{Q}\,z\,,$$

$$\mathrm{Var}\left(\phi(y)\,\middle|\,\phi(x) = \sqrt{Q}\,z\right) = \left(1 - F\left(1 - \tfrac{2h_n}{n}\right)^2\right)Q\,, \tag{18}$$

so that

$$P_n(a, z) = \mathbb{P}\left(\phi(y) < 0\,\middle|\,\phi(x) = \sqrt{Q}\,z\right) = \Phi\left(-\frac{F\left(1 - \tfrac{2h_n}{n}\right)z}{\sqrt{1 - F\left(1 - \tfrac{2h_n}{n}\right)^2}}\right)$$

$$= \Phi\left(-\frac{z}{2}\sqrt{\frac{n}{F'(1)\,h_n}}\left(1 + O\left(\sqrt{\tfrac{h_n}{n}}\right)\right)\right)\,, \tag{19}$$

where

$$\Phi(t) = \int_{-\infty}^{t} e^{-\frac{s^2}{2}}\,\frac{\mathrm{d}s}{\sqrt{2\pi}} \tag{20}$$

and we have used (14). Using that $\ln\Phi(-t) = -\frac{t^2}{2} - \frac{1}{2}\ln(2\pi t^2) + O(\frac{1}{t^2})$ for $t \to \infty$ we get

$$\ln P_n(a, z) = -\frac{n\,z^2}{8F'(1)h_n} + O\left(\sqrt{\tfrac{n}{h_n}}\right) = -\frac{z^2\sqrt{n\ln n}}{8F'(1)a} + O\left(\sqrt[4]{n\ln n}\right)\,. \tag{21}$$

We have

$$N_n(a, z) = \binom{n}{h_n} P_n(a, z)\,. \tag{22}$$

Using that $\ln k! = \left(k + \frac{1}{2}\right)\ln k - k + O(1)$ for $k \to \infty$ we get

$$\ln\binom{n}{h_n} = h_n\left(\ln\frac{n}{h_n} + 1\right) - \frac{1}{2}\ln h_n + O(1) = \frac{a}{2}\sqrt{n\ln n} + \frac{a}{2}\sqrt{\frac{n}{\ln n}}\ln\frac{\ln n}{a^2} + O(\ln n)\,, \tag{23}$$

and the claim follows.

## 3 Proof of Theorem 2

Let $\varphi : [0, 1] \to \mathbb{R}$ be a random function with a Gaussian probability distribution such that for any $s, t \in [0, 1]$

$$\mathbb{E}(\varphi(t)) = 0\,, \quad \mathbb{E}(\varphi(s)\,\varphi(t)) = F(1 - 2\,|s - t|)\,. \tag{24}$$

From (24), for any $s, t \in [0, 1]$, $\varphi(s) - \varphi(t)$ is a Gaussian random variable with zero average and variance

$$\mathbb{E}\left((\varphi(s) - \varphi(t))^2\right) = 2 - 2F(1 - 2\,|s - t|)\,. \tag{25}$$

Recalling that $F(1) = 1$, there exists $\epsilon > 0$ such that for any $0 \le u \le 2\epsilon$ we have $1 - F(1 - u) \le (F'(1) + 1)u$. Hence, if $|s - t| \le \epsilon$ we have

$$\mathbb{E}\left((\varphi(s) - \varphi(t))^4\right) = 12\left(1 - F(1 - 2\,|s - t|)\right)^2 \le 48\left(F'(1) + 1\right)^2 |s - t|^2\,. \tag{26}$$

Then, the Kolmogorov continuity theorem [4] implies that with probability one the function $\varphi$ is continuous. Let $t(\varphi)$ be the minimum $0 \le t \le 1$ such that $\varphi(t) = 0$:

$$t(\varphi) = \min\left\{\inf\left\{0 \le t \le 1 : \varphi(t) = 0\right\}, 1\right\}\,. \tag{27}$$

Since with probability one $\varphi$ is continuous and $\varphi(0) \neq 0$, we have $\varphi(t) \neq 0$ in a neighborhood of $0$, hence $t(\varphi) > 0$ with probability one. Therefore, the expectation value of $t(\varphi)$ is strictly positive: $t_0 = \mathbb{E}(t(\varphi)) > 0$.

From (8), for any $i, j = 0, \ldots, n$ we have $\mathbb{E}(\phi(x^{(i)})) = 0$ and

$$\mathbb{E}\left(\phi\left(x^{(i)}\right)\phi\left(x^{(j)}\right)\right) = Q\,F\left(1 - \tfrac{2|i - j|}{n}\right)\,. \tag{28}$$

Comparing with (24) we get that $\{\phi(x^{(i)})\}_{i=0}^n$ have the same probability distribution as $\{\sqrt{Q}\,\varphi(\frac{i}{n})\}_{i=0}^n$. From the definition of $t(\varphi)$, for any $1 \le i < n\,t(\varphi)$, $\varphi(\frac{i}{n})$ has the same sign as $\varphi(0)$. Therefore, $h_n \ge n\,t_0$, and the claim follows.

Table 1

| Number of input bits | Search method | % of points at distance | | | | |
|---|---|---|---|---|---|---|
| | | 1 | 2 | 3 | 4 | 5+ |
| **50** | Exhaustive | 45.4% | 28.0% | 14.6% | 6.7% | 5.3% |
| | Greedy | 45.2% | 28.8% | 14.3% | 6.4% | 5.3% |
| **100** | Exhaustive | 38.3% | 27.1% | 15.9% | 9.8% | 8.9% |
| | Greedy | 35.8% | 26.7% | 15.6% | 10.8% | 11.1% |
| **150** | Exhaustive | 29.1% | 26.3% | 17.9% | 12.0% | 14.7% |
| | Greedy | 31.6% | 22.6% | 18.2% | 11.2% | 16.4% |

## 4   Experiments on random deep neural networks

Table 1 shows Hamming distances of random bit strings to the nearest differently classified bit string measured using a heuristic greedy search algorithm and an exact search algorithm. Resulting breakdowns for the two algorithms are consistent across all network input sizes tested. For each algorithm and network input size, Hamming distances to nearest differently classified bit strings from a random bit string were evaluated 1000 times with each evaluation performed on a randomly created neural network.