[Reviews · NeurIPS 2019]

Reviewer 1



The manuscript is clear, original, well written, and contributes towards the theoretical understanding of an important problem. The only weakness appears to be the overreliance on the Guassian assumption for the features and the assumed regularity of the covariance function, which is satisfied by ReLU but not other activation functions, such as tanh.

Reviewer 2



Summary: If we assume the inputs of DNNs to be binary (e.g. black-white MNIST images), then the authors show that the Hamming distance between data points (e.g. the number of pixels that need to be flipped) with two different classes grows at least a \sqrt{N / log(N)} rate. The proof utilizes the (now standard) Gaussian process approximation results for DNNs. This theoretical result suggests that DNNs produce functions with low Kolmogorov complexity, which is useful for studying generalization bounds of DNNs. Some experiments on random data and on tiny nets on MNIST (in the supplement) are presented, empirically verifying the bounds. I tend to weakly reject this paper due to the weakness of the empirical and theoretical results, and on the organization of the paper (MNIST results in main text?). Given result 2), I’d strongly suggest that the authors think about what might be possible from an adversarial setting. Connecting the fact that the average number of features changed to change a classification seems like a potential way to provide some sort of guarantee against adversarial attacks. This could provide another alternative theoretical result if done properly. Originality: This paper seems to be a natural theoretical extension of the work of [27]. In particular, the authors of that paper compute the marginal likelihood of the GP representation of DNNs and derive a generalization bound based of that, with extensive empirical results. Similarly, they also hypothesize and empirically study the string complexity of small functions like is studied here, empirically finding that lower complexity functions are produced with higher probability. As such, I don’t think that the theoretical results and limited empirical results (only on MNIST and toy data) are significant or original enough for a full paper at this time. I’d strongly suggest utilizing the Hamming distance bound proved in this paper in an experimental set-up like that of Section 4 and Appendix F of [27]. Particularly, the results in Figure 1 ought to be able to be replicated on MNIST and small portions of (binarized?) CIFAR for the Hamming distance type bounds shown in this paper. Clarity: Overall, the paper is mostly nice to read and free of grammatical errors. One major organization issue that I noticed is that the MNIST results are heavily mentioned in the abstract and conclusion, but are nowhere in the main text. They are instead only found in the Appendix. This is somewhat misleading, as a reader of the paper might be confused and not see them. I’d strongly suggest that these results be moved into the main text – either by swapping out the toy problem experiments or by moving one of the detailed proofs of Theorem 1 or 3 to the Appendix instead. Lines 73-82: “A fundamental consequence … provided by the Kolmogorov complexity.” This sentence and the rest of the paragraph could be rephrased as “DNNs produce a class of functions that are a subset of all functions with low Kolmogorov complexity.” The rest is unclear logically and somewhat pedantic. Lines 54-82: This paragraph seems to be run-on. To further emphasize the contributions, it would be nicer to have bullet points, or at least split paragraphs here. Significance: To make this quite interesting from a theoretical perspective, can the Hamming distance be shown to a) align with some sort of PAC Bayes bound and/or b) empirically correlate with generalization error? That is, can you make the connections between simplicity of functions and generalization error rigorous from a learning theoretic perspective? It’s possible that this is somewhere in the references, but it should be specifically mentioned in the main text. Quality: Lines 135-142: Assuming that the inputs “l[ie] on a sphere” is not quite the same assuming that the inputs are binarized on a grid. Additionally, the binarized on a grid assumption seems like it would only hold for binary images. How would the authors expand this to either tabular (continuous) data on a sphere or image data [0, 1]^d x [0, 1]^d x [0, 1]^d (standard color images)? Line 150: “from the central limit theorem, [the post-activation] has a Gaussian distribution” While this assumption is standard in the literature [39, 41, 42, and others], the rate of convergence here could be arbitrarily small (we’d actually expect the post-activations to have much heavier tails, see Vladmirova, et al, 2019). This could make the rate of convergence quite slow, and ought to be mentioned. Appendix: In my understanding, the relationship between Hamming distances on random data, the MNIST training data, and the MNIST test data is quite under-explored. While the proofs in the paper seem to be interesting mathematically, converting these bounds into practically useful tools seems to require the inflation on real data. Why do the authors think that the Hamming distances on the real image datasets are inflated? Line 152: Could the authors explain what the understanding of the function F(.) is throughout the proof? From skimming the supplementary material, it seems intimately related to the basis expansion of the kernel corresponding to the activation function. If this is the explanation, a sentence explaining this would seriously help in understanding the proofs. References: Vladmirova, et al, 2019. Understanding Priors in Bayesian Neural Networks at the Unit Level, ICML, http://proceedings.mlr.press/v97/vladimirova19a/vladimirova19a.pdf --------- Post-Rebuttal I'm inclined to raise my score to a weak accept as the authors have convinced me that a) they will move the MNIST results to the main text as necessary and b) that the bounds on the Hamming distance can be extended to different data types. My concern of the potential weakness of this work in comparison to [27] has also been alleviated somewhat by the other reviewers. I'd also like to thank the authors for providing a further experiment demonstrating that the Hamming distance computation seems to be empirically interesting on real datasets. Also, please adjust the citation of [27] to include the OpenReview link (the published ICLR version): https://openreview.net/forum?id=rye4g3AqFm .

Reviewer 3



After reading through the author response, the authors have not only agreed to make the organizational changes requested but also added a new experiment. We maintain that this paper should be accepted. We had one reviewer disagree on the basis that this is an incremental theoretical result from [27], however the theoretical result of this paper is a new one that expands on just one of the empirical observations made in [27]. The theoretical result here is still a significant one, in our opinion. -------------------------- Before Rebuttal The authors present a novel proof regarding the Hamming distance of random bit string inputs with different binary classifications to random neural networks. Their proof helps explain some experimental results observed in prior work [27] and notably, their predicted theoretical results are experimentally confirmed even after a network has been trained. The mathematical set up of Gaussian process approximation for their proof is by the author’s admission not novel, coming from [41]. However, they expand on this setup to show that classification outputs of nearby (short-distance) inputs are almost perfectly correlated, i.e. they have the same classification. This expansion on [41] allows them to construct their proof of Theorem 1. The major drawbacks of this paper are due to presentation style and organization. The immediately obvious mistake is that the authors label Theorems 1 and 3, but there is no Theorem 2. More importantly, the paper requires the reader to bounce between the proof logic in the body of the paper and the proof setup in the supplement. The result on MNIST is also buried in the supplement. While it is understandable that the proof setup must stay in the supplement due to space restrictions, it might be more prudent to move the heuristic argument into the supplement and the MNIST result into the main paper. Another option is to move the proof entirely into the supplement, leaving only the theorem statements, the heuristic argument for intuition, and all experimental results in the main paper. Section 1.1, while very clear in explaining how the authors’ work fits into the research landscape, could at least be broken into paragraphs and made a bit more succinct. It would also be nice to see some standard deviation error bars in the figures. The supplement also mentions the greedy algorithm for finding a nearest neighbor bit string as being in Section 6.1 of the main paper, when there is no Section 6 in the main paper.

[Author Response · NeurIPS 2019]

We thank all the Reviewers for a careful reading of our paper and for providing useful suggestions for improvements, which we will be happy to implement in the camera-ready version.

## Reviewer #1

As we state at the beginning of Sec. 2, Theorems 1 and 3 hold for any activation function, including $\tanh$. Theorem 1 states that in the limit $n \to \infty$ of infinite length of the bit string with the number of layers $L$ kept fixed the Hamming distance scales as $\sqrt{n/(F'(1)\ln n)}$, where $F'(1)$ depends on $L$ but not on $n$. As we state in Remark 2, for ReLU we always have $F'(1) \leq 1$, while for $\tanh$ depending on the variances of weights and biases $F'(1)$ may grow exponentially with $L$. Therefore, for finite values of $L$ and $n$ with the $\tanh$ activation function, $F'(1)$ may become comparable with $n/\ln n$ and significantly affect the Hamming distance. We will clarify this point in the camera-ready version.

## Reviewer #2

Exploiting our results to understand the stability of trained neural networks under adversarial perturbations is an extremely interesting line of research which we are currently pursuing.

We have performed additional experiments on the MNIST dataset to explore the correlation between the Hamming distance of a training or test picture from the closest classification boundary and the correctness of its classification. Figure 1 shows that incorrectly classified pictures are significantly closer to the boundary than correctly classified ones, thus implying an empirical correlation between Hamming distance and generalization. We will include a discussion in the camera-ready version.

Figure 1: Histogram of correctly and incorrectly classified pictures shows that trained neural networks are far more likely to misclassify points closer to a classification boundary for both the training and test sets. Results are aggregated across 20 different trained neural networks. All neural networks are trained to classify whether digits are even or odd and are trained for 10 epochs using the adam optimizer on the MNIST dataset.

We will move the MNIST results to the main paper swapping them with the detailed proofs and modify Sec. 1.1 as suggested.

The kernel entry associated to two inputs lying on the sphere is a function of their squared Euclidean distance, which coincides with the Hamming distance in the case of bit strings. We are currently working on extending our results to continuous inputs replacing the Hamming distance with the squared Euclidean distance.

We will add in the camera-ready version a discussion on the convergence rate to the Gaussian probability distribution.

We conjecture that the training process keeps the classification boundaries as far as possible from the training pictures, and this results in having most of the pictures that represent a digit still far from the boundaries. Therefore, the distance to the closest boundary is larger for a training or test picture than for a random picture. We will add a comment on this in the camera-ready version.

$F$ is the function that provides the entries of the kernel of the Gaussian process associated to the neural network in terms of the scalar product of the inputs, as defined in eq. (2). We will make the definition more clear in the camera-ready version.

## Reviewer #3

We will implement all the suggestions: we will relabel Theorem 3 as Theorem 2, move the MNIST analysis to the main paper swapping it with the detailed proofs, modify Sec. 1.1 as suggested and add error bars to the plots. The reference to Sec. 6.1 is a typo due to a previous version of the paper, the correct reference is Sec. 4.1.

[Meta-Review · NeurIPS 2019]

The reviewers agreed that this was a very interesting submission, well-written and elegant, with a significant theoretical advance in our understanding of the effectiveness of neural networks. This advance nicely builds on previous empirical work by bringing theory to bear on explaining phenomena previously only demonstrated experimentally.